# Systematic mapping of protein-metabolite interactions in central metabolism of *Escherichia coli*

Maren Diether[1,2,†], Yaroslav Nikolaev[3,†,*] 🆔, Frédéric HT Allain[3] 🆔 & Uwe Sauer[1,**] 🆔

## Abstract

Metabolite binding to proteins regulates nearly all cellular processes, but our knowledge of these interactions originates primarily from empirical *in vitro* studies. Here, we report the first systematic study of interactions between water-soluble proteins and polar metabolites in an entire biological subnetwork. To test the depth of our current knowledge, we chose to investigate the well-characterized *Escherichia coli* central metabolism. Using ligand-detected NMR, we assayed 29 enzymes towards binding events with 55 intracellular metabolites. Focusing on high-confidence interactions at a false-positive rate of 5%, we detected 98 interactions, among which purine nucleotides accounted for one-third, while 50% of all metabolites did not interact with any enzyme. In contrast, only five enzymes did not exhibit any metabolite binding and some interacted with up to 11 metabolites. About 40% of the interacting metabolites were predicted to be allosteric effectors based on low chemical similarity to their target's reactants. For five of the eight tested interactions, *in vitro* assays confirmed novel regulatory functions, including ATP and GTP inhibition of the first pentose phosphate pathway enzyme. With 76 new candidate regulatory interactions that have not been reported previously, we essentially doubled the number of known interactions, indicating that the presently available information about protein–metabolite interactions may only be the tip of the iceberg.

**Keywords** allostery; central metabolism; nuclear magnetic resonance; protein–metabolite interactions; regulation
**Subject Categories** Metabolism; Methods & Resources
**Mol Syst Biol. (2019) 15: e9008**

## Introduction

Robustness and adaptability of cells emerge from the dynamic regulatory interplay of diverse biomolecules. Interactions with metabolites are of particular importance as information input to many regulatory proteins such as transcription factors and kinases (Li *et al*, 2010; Hahn & Young, 2011; Kochanowski *et al*, 2013, 2017; Chubukov *et al*, 2014; Wegner *et al*, 2015), and also into the metabolic reaction network itself (Gerosa & Sauer, 2011; Ljungdahl & Daignan-Fornier, 2012). An extensive network of protein–metabolite interactions enables coordination of the various activities in each cell. Our knowledge about these functional interactions is largely based on accumulated biochemical evidence from studies on individual proteins that typically yield a few new interactions by testing compounds on the basis of existing knowledge (Jones & Fink, 1982; Cherry *et al*, 2012; Keseler *et al*, 2017; Placzek *et al*, 2017).

While great strides have been made toward systematic mapping of physical protein–protein and protein–DNA interaction networks (Cai & Huang, 2012; Syafrizayanti *et al*, 2014; Myers *et al*, 2015; Smits & Vermeulen, 2016), systematic mapping of protein–metabolite interactions is lagging behind. One challenge is the generally low affinity (mM range) of protein–metabolite interactions (Reznik *et al*, 2017) and their fleeting nature. Some large-scale discovery approaches reported hundreds of novel protein–metabolite interactions with nearly no overlap between the studies (Gallego *et al*, 2010; Savitski *et al*, 2014; Geer & Fitzgerald, 2016; Piazza *et al*, 2018), providing a glimpse on the size of the interaction space. Given this vast space, it might be more desirable to systematically map protein–metabolite interactions within a defined subnetwork. While this has been achieved for non-polar metabolites (lipids; Gallego *et al*, 2010; Li *et al*, 2010), these methods are not transferable to polar metabolites. Alternative methods suitable for polar metabolites suffer from other problems. For example, most available high-throughput methods do not detect metabolite binding itself, but instead measure indirect interaction consequences on the protein (Savitski *et al*, 2014; Diether & Sauer, 2017; Piazza *et al*, 2018), thus missing interactions that elicit only weak effects or are

1  Institute of Molecular Systems Biology, ETH Zurich, Zurich, Switzerland
2  Life Science Zurich PhD Program on Systems Biology, Zurich, Switzerland
3  Institute of Molecular Biology and Biophysics, ETH Zurich, Zurich, Switzerland
   *Corresponding author. Tel: +41 44 633 0720; E-mail: yaroslav.v.nikolaev@gmail.com
   **Corresponding author. Tel: +41 44 633 36 72; E-mail: sauer@imsb.biol.ethz.ch
   † These authors contributed equally to this work

condition-dependent. The remaining methods are hampered by chemical limitations, requiring functionalization (Hulce et al, 2013; Höglinger et al, 2017) or radiolabeling (Roelofs et al, 2011) of metabolites, or high protein concentrations (Orsak et al, 2012). To address these limitations, we recently showcased a nuclear magnetic resonance (NMR) spectroscopy approach that permits direct detection of interactions between any set of water-soluble proteins and metabolites (Nikolaev et al, 2016). In a proof-of-concept study with four proteins and 33 metabolites, we recovered all known and detected new interactions, some of which proved to be functional modulators. This approach thus opened a venue to exhaustively search a pre-selected space of proteins and metabolites for potential interactions.

To elucidate whether our current knowledge on protein–metabolite interactions is nearing completion, at least for well-characterized subnetworks, we chose to investigate Escherichia coli central carbon metabolism that has been thoroughly investigated over decades. At present, about 100 regulatory (metabolite changes enzyme activity) and 130 catalytic (metabolite is substrate or product) interactions involving the 35 major isoenzymes of central metabolism are reported in the EcoCyc database (Keseler et al, 2017). We systematically generated ligand-detected NMR interaction profiles of 29 purified enzymes from E. coli central metabolism with 55 selected metabolites, between which 72 interactions were already known. Here, we focused our analysis only on high-confidence NMR interactions by choosing a false-positive rate cutoff of 5%, which yielded a dataset encompassing 30% of the 72 known interactions. At the above cutoff, we detected 98 interactions between all tested enzymes and metabolites, including 22 known interactions and 76 interactions that had not been reported previously, and validated five of the newly predicted interactions with in vitro enzyme assays. Among the most striking observations was the highly promiscuous binding of GTP and other purine nucleotides (ATP, AMP, and cAMP), and the lack of interactions with metabolites from amino acid biosynthesis.

## Results

### Ligand-detected T1rho NMR assay for a biological subnetwork

To probe the depth of our current knowledge on protein–metabolite interactions in E. coli central metabolism, we selected all monomeric and homo-oligomeric enzymes. Hetero-oligomeric and membrane-bound proteins were excluded because of expected difficulties with purification and in vitro reconstitution. For reactions catalyzed by more than one enzyme, the major isoenzyme was chosen. The resulting 35 selected central metabolic enzymes were purified by His-tag affinity purification from clones of the ASKA library (Kitagawa et al, 2005). For six of the enzymes (AceB, GltA, Ppc, PpsA, PrpC, and SthA), we achieved only low yields under high-throughput purification conditions, reducing the final set to 29 enzymes (Dataset EV1).

For systematic testing of putative regulators, we selected 59 metabolites from several pathways, including amino acids, nucleotides, cofactors, and central metabolism, several with known regulatory functions (Dataset EV1). From the initial set of 59, four metabolites (coenzyme A, acetyl-coenzyme A,

erythrose-4-phosphate, and 5-aminoimidazole-4-carboxamide ribonucleotide) were not tested due to their instability in our buffer and temperature conditions. To detect protein–metabolite interactions, purified proteins were mixed with a subset of metabolites and NMR spectra were recorded. A single one-dimensional (1D) NMR spectrum can resolve few dozens of individual metabolite signals. Due to differences in the NMR properties of small and large molecules, metabolite signals broaden (exhibit reduced intensity) upon protein binding. We exploit this change in signal intensity to detect metabolite–protein interactions. High-throughput NMR analysis relies on availability of isolated compound-specific peaks in the combined NMR spectrum of specific metabolite mixtures. NMR signals start overlapping as the complexity of the metabolite mixture increases, thus limiting the number of metabolites that can be confidently assayed within one mixture. To split the selected metabolites into a minimum number of groups (mixes), the NMRmix tool (Stark et al, 2016) was employed. Three metabolites were explicitly assigned to different mixes to avoid potential enzymatic reactions, yielding four mixes each containing 12, 14, 14, and 15 metabolites in such a way that each metabolite had at least one well-separated signal in the 1D $^1$H NMR spectrum of the mix (Fig 1A and B, Appendix Figs S1–S4, Dataset EV1).

To automate data acquisition, we developed a set of Python-based TopSpin libraries for sample changing, basic experiment setup, and spectral processing. To increase the measurement throughput compared to the pilot study (Nikolaev et al, 2016), we only measured 1D hydrogen-detected ($^1$H) T1rho relaxation spectra and omitted water-ligand observed via gradient spectroscopy (WaterLOGSY), as the former appeared more robust although slightly less sensitive (Nikolaev et al, 2016). T1rho experiments detect signal decay rates (relaxation rates) in metabolites upon binding to a large macromolecule target. If an interaction between metabolite and target exists, the NMR signals of the metabolite decay (disappear) faster. To minimize contributions of metabolite instability to the signals in the final spectra, T1rho spectra with short relaxation delay (10 ms) were measured as two identical repetitions, before and after the T1rho experiment with long relaxation delay (200 ms), and summed up during processing. Furthermore, the difference spectrum of the two short-delay T1rho spectra provided a quality filter to detect metabolites showing increased chemical instability in the presence of specific proteins. Among other sources of instability, metabolite degradation could be the result of enzymatic conversion, although this is not likely to be a major confounding factor given that the protein–metabolite mixture was incubated for several hours prior to NMR recording. However, differentiating the various sources of metabolite instability is not feasible given our current setup. To identify protein–metabolite interactions, purified proteins were mixed individually with the four metabolite mixes in excess (15 μM of protein monomer and 200 μM of each metabolite) in a buffer optimized for physiological salt concentrations (Fig 2). NMR measurements with pure enzymes and four pure metabolite mixes were used as references for quantification.

Interactions between enzymes and metabolites were analyzed using a fully automated custom-built analysis pipeline, initially by computing both the fractional signal intensity (Nikolaev et al, 2016) and relaxation factor (ΔRF; Gossert & Jahnke, 2016). For the final analyses, the ΔRF metric was selected as it gave slightly better

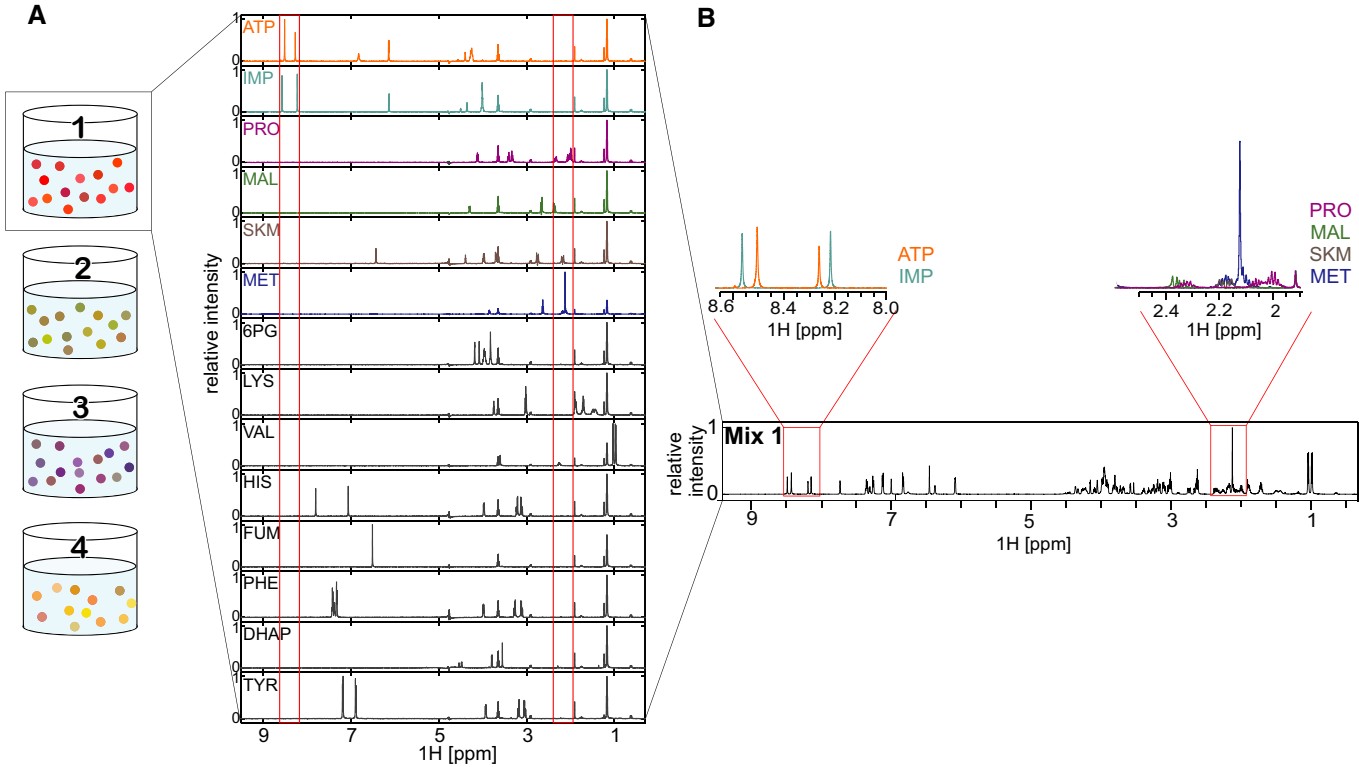

**Figure 1. NMR spectra of metabolite mixes.**

A 1D ¹H-NMR spectra of a metabolite mix and the individual metabolites contained therein.

B Identification of single compound peaks from 1D ¹H-NMR spectra of a metabolite mix. Compound detection is exemplified by showing sections of the 1D ¹H-NMR spectra of ATP, IMP, proline (PRO), malate (MAL), shikimate (SKM), and methionine (MET).

results for the given set of proteins and metabolites (assessed by comparing the "area under the curve" for both metrics; Appendix Fig S5). The ΔRF metric quantifies the difference in the signal relaxation rate of the metabolite alone and in the presence of the protein. If an interaction between metabolite and target exists, the difference ΔRF value increases above zero.

$$\Delta RF = RF_M - RF_{PM} = \left(\frac{M_{200ms}}{M_{10ms}}\right) - \left(\frac{(PM)_{200ms} - P_{200ms}}{(PM)_{10ms} - P_{10ms}}\right)$$

Under the NMR setup employed, interactions with dissociation constants ($K_D$s) in the μM-to-mM range were generally detectable, with μM $K_D$s producing the strongest attenuation of metabolite signals in the presence of the protein, i.e., highest ΔRF (Gossert & Jahnke, 2016; Nikolaev *et al*, 2016). The final analysis used here was restricted to peaks with a signal-to-noise ratio greater than two in the final T1rho NMR spectra of the protein–metabolite mixtures (after subtracting pure protein and metabolite signals).

**Systematic map of protein–metabolite interactions in *Escherichia coli* central metabolism**

To determine the ΔRF values that represent biologically relevant interactions, we investigated the overlap of the interactions detected by our approach with the known interactions reported in the EcoCyc database (Keseler *et al*, 2017). Among the 29 enzymes and 55 metabolites, 43 catalytic (metabolite is substrate or product) and 40 regulatory (metabolite changes enzyme activity) interactions are known. Since some regulatory metabolites are also substrates or products of their target enzyme, we thus have 72 previously known interactions. Using different ΔRF cutoffs, we calculated the false-positive and true-positive rates of recovering the known interactions, obtaining a receiver-operator characteristics curve (Appendix Fig S5, Materials and Methods: Analysis of the recovery of known interactions). For conservative discovery—to select only high-confidence interactions—we chose a false-positive rate of 5%, corresponding to a ΔRF cutoff of 0.1805 that was applied in all subsequent analyses.

By applying the 5% false-positive rate cutoff, we detected 98 distinct protein–metabolite interactions in our dataset. We recovered 22 of the 72 previously reported interactions (30%), thus achieving a twofold higher true-positive rate than a recent MS-based study (Piazza *et al*, 2018; 16% recovery of known interactions). Half of the 55 tested metabolites did not exhibit any interaction and the other half interacted on average with 3.5 proteins each (Fig 3A, Dataset EV2). Remarkably, only four of the 20 proteinogenic amino acids (Asn, His, Leu, and Trp) exhibited interactions with the tested enzymes in our assays. In contrast, the ten tested nucleotide-related metabolites interacted with 36 enzymes, most prominently GTP

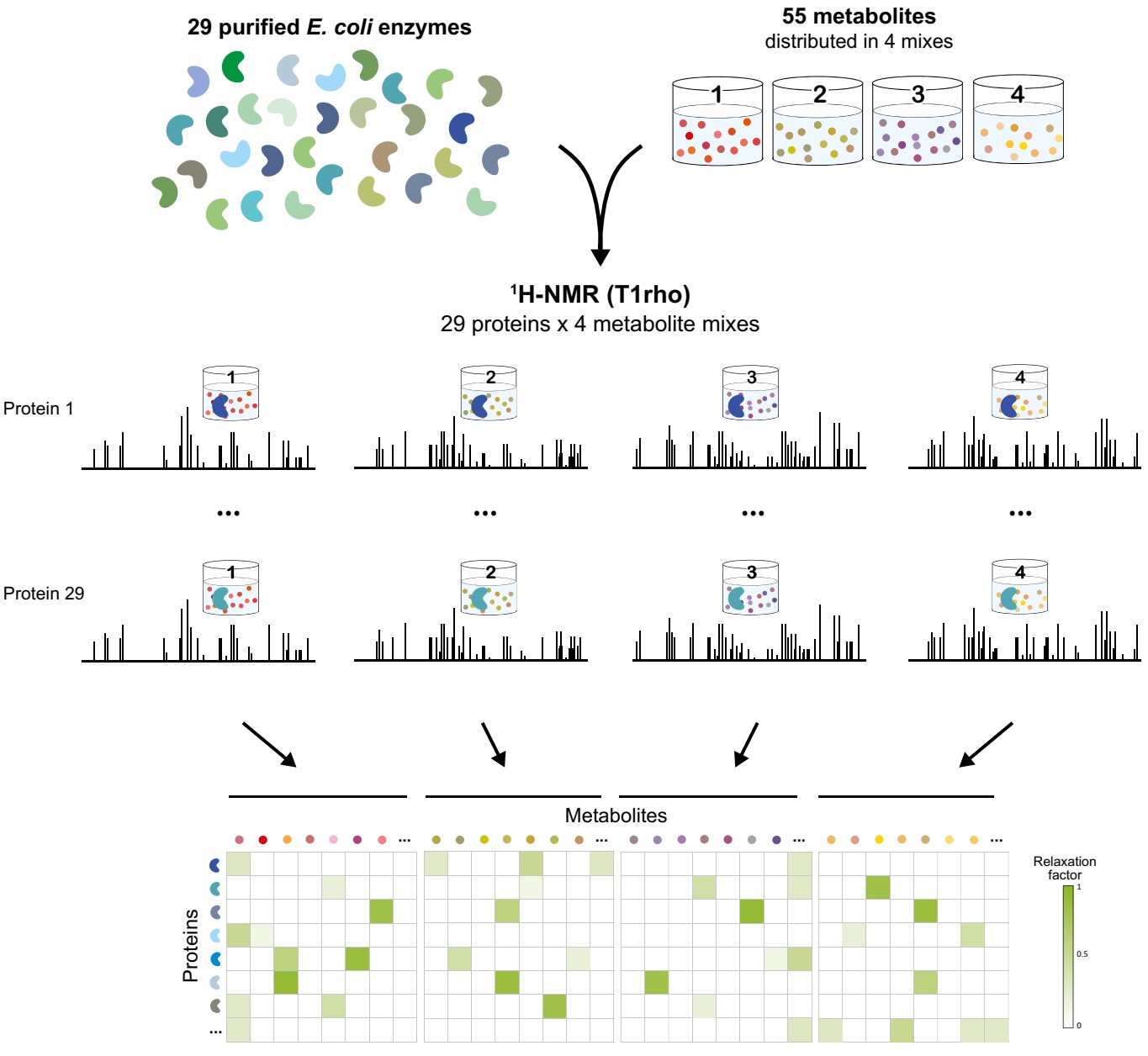

**Figure 2. Workflow of ligand-detected NMR approach.**

Twenty-nine His-tagged enzymes of *Escherichia coli* central metabolism were mixed individually with four metabolite mixes, and T1rho 1D $^1$H NMR spectra were recorded for every protein–mix combination. All possible interactions between enzymes and metabolites were quantified using the relaxation factor and are displayed in a protein–metabolite interaction map.

with 13 and AMP with nine proteins. Overall, the here-detected interactions were equally distributed between regulatory and catalytic interactions (Fig 3B), indicating that our NMR approach is not biased by interaction type. Likewise, there does not appear to be a bias through chemical structures as the interactions spanned a wide range of metabolites, as expected from NMR T1rho relaxation experiments (Hajduk *et al*, 1997). Only five tested proteins did not interact with any metabolite, and the remaining 24 proteins interacted with about four metabolites on average (Appendix Fig S6). The most highly connected enzymes were fructose-bisphosphate aldolase

(FbaA) and malate dehydrogenase (MaeB) with eleven interacting metabolites each. Projection of the newly detected interactions onto the network of central metabolism revealed significantly fewer interactions in the tricarboxylic acid (TCA) cycle (*P*-value 0.004, two-tailed *t*-test assuming unequal variance; on average 1 per TCA cycle enzyme vs. 3.2 for any other enzyme, excluding purely catalytic interactions; Fig 3C). We observed no significant differences between enzymes that catalyze reversible or irreversible reactions (*P*-value 0.22, two-tailed *t*-test assuming unequal variance). The number of interacting metabolites per protein did not correlate with

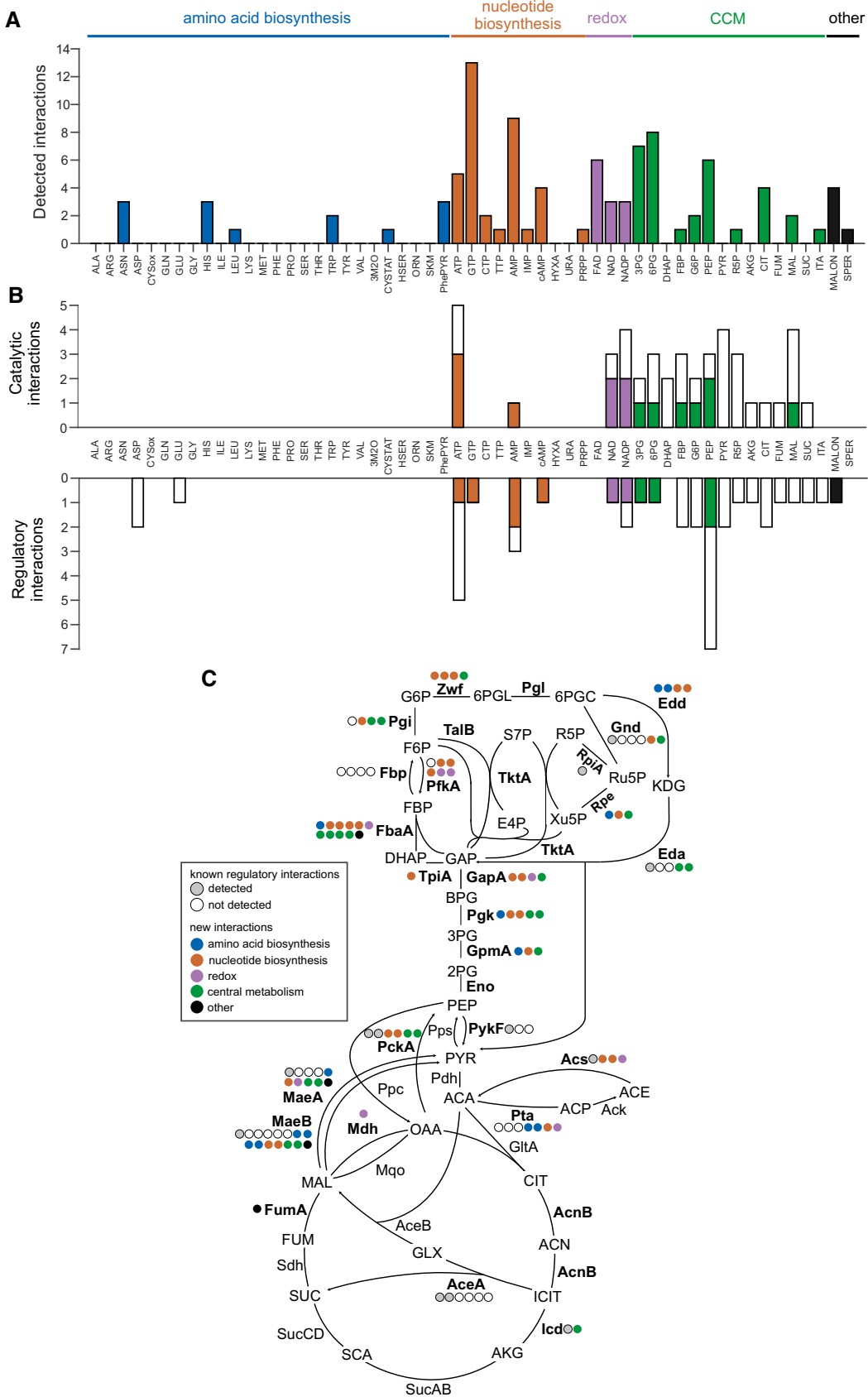

**Figure 3.**

**Figure 3. Overview of enzyme–metabolite interactions detected with NMR.**

A   Number of interactions detected per metabolite. Metabolites are grouped according to biological pathways; the height of the bar indicates the total number of interactions.

B   Recovery of interactions reported in EcoCyc database. Metabolites are grouped according to biological pathways, the total height of the bar indicates the number of known interactions that could have been detected, and the height of the colored bar indicates the number of actually recovered interactions. Recovery of catalytic (metabolite is substrate or product of the enzyme) and regulatory interactions is shown.

C   Distribution of known regulatory and newly predicted interactions in *Escherichia coli* central metabolism. Enzymes included in this study are depicted in bold. Gray and white circles indicate how many known interactions were recovered or not recovered, respectively. Newly detected interactions are depicted using circles that are color-coded according to the origin of the respective metabolite.

Data information: Abbreviations of proteins and metabolites are explained in Dataset EV1.

protein size (Appendix Fig S7), suggesting that experimental results are not biased by the molecular weight of the targets. Sequence-level analysis (Gasteiger *et al*, 2005) showed no correlation of protein aliphatic content and hydropathicity with the number of significant hits observed ($R^2 = 0.0009$ and $0.0007$; Appendix Fig S8). Similarly, metabolite hydrophobicity did not correlate with the number of detected interactions ($R^2 = 0.0844$; Appendix Fig S8). In total, we discovered 76 new protein–metabolite interactions.

**Chemical similarities distinguish between potential allosteric and competitive interactions**

In general, all identified enzyme–metabolite interactions represent potential catalytic and/or regulatory interactions. For catalytic interactions, binding must occur at the enzymes' active site. For regulatory interactions, binding can occur either at the active site (competitive regulation) or at an alternative binding site (allosteric regulation). Such allosteric regulators often have stronger regulatory potential, as their effect generally does not depend much on the concentrations of the enzymes' native substrates [e.g., in the case of non-competitive inhibition (Purich, 2010)]. To differentiate binding modes, we investigated the chemical similarity of metabolites with substrates and products of the tested enzymes. We assume that competitive binders will have a higher similarity to substrates or products than allosteric binders. To investigate the underlying distribution of chemical similarities in our biological subnetwork, we calculated the maximum global chemical similarity between all possible regulator–substrate/product pairs using Simcomp2 (Hattori *et al*, 2010; Fig 4). Simcomp2 identifies the maximal common substructure of two chemical structures using a graph-based method (Hattori *et al*, 2010). The resulting distribution ranges from zero (no similarity) to one (perfect similarity; regulator is identical to substrate or product) with a mean of 0.34. Computing this metric for the 98 detected interactions results in a distribution with a mean of 0.59 (Fig 4, Dataset EV2), implying that most NMR-detected interactors are similar (> 0.5) to the natural substrates/products of their target. In turn, this indicates that many of the detected interactions are due to binding of the metabolite to the enzyme active site. Nevertheless, 40% of the NMR-detected interactors have a low chemical similarity (< 0.5) to substrates/products of the target enzyme, suggesting an allosteric binding mode (Fig 4B). Since they are more distant from central metabolism, amino acid and nucleotide interactors expectedly dominate among the putative allosteric binders (Fig 4A, black rectangles). Overall, we predict that 36 out of the 76 newly discovered interactions are allosteric while the remaining interactions have a competitive binding mode (Appendix Fig S9).

**Validation of newly predicted protein–metabolite interactions with *in vitro* enzyme assays**

Given the established reliability of NMR in detecting molecular interactions (Dalvit *et al*, 2006; Pellecchia *et al*, 2008), we decided to directly validate the newly predicted interactions at the functional level. To validate the functionality of the predictions, we chose eight enzyme–metabolite pairs based on their biological relevance and chemical similarity and tested the enzyme activities by *in vitro* assays in the presence and absence of the predicted regulator (summarized in Appendix Fig S10, Dataset EV3). First, the $NADP^+$-dependent glucose-6-phosphate dehydrogenase (Zwf) was selected for being at the branch point between glycolysis and pentose phosphate pathway. Our NMR-based approach predicted interactions of Zwf with the nucleotides ATP, GTP, and IMP, with chemical similarities to the natural substrates/products of 0.60, 0.55, and 0.44, respectively. These interactions were not reported in EcoCyc or BRENDA, except for ATP inhibition of Zwf in the absence of stabilizing $Mg^{2+}$ ions (Santimoy Banerjee & Fraenkel, 1972). In our plate-reader-based *in vitro* enzyme assay, ATP and GTP indeed inhibited Zwf even at physiological concentrations of $MgCl_2$. IMP did not affect Zwf activity, similar to the negative control AMP (Fig 5A). Second, we investigated the predicted regulation of phosphate acetyltransferase (Pta) by L-tryptophan and phenylpyruvate. These interactions were selected due to the very low similarity between the regulators and the natural substrates and products of the enzyme (0.06 and 0.05), indicating possible allosteric interactions. Mass spectrometry-based *in vitro* assays showed a small, but insignificant concentration-dependent inhibitory effect of phenylpyruvate and no inhibitory effect of L-tryptophan on phosphate acetyltransferase activity (Fig 5B). Third, the glycolytic fructose-bisphosphate aldolase class II (FbaA) has no reported metabolite regulators in EcoCyc but was found to interact with eleven metabolites. We selected 3-phosphoglycerate, ATP, and phosphoenolpyruvate for validation, as well as hypoxanthine and IMP as negative controls. Mass spectrometry-based *in vitro* assays showed that 3-phosphoglycerate, ATP, and phosphoenolpyruvate had an activating effect on the enzyme, whereas hypoxanthine and IMP had no measurable impact (Fig 5C). Previously, 3-phosphoglycerate has been reported to inhibit fructose-bisphosphate aldolase (Szwergold *et al*, 1995) and the activating effect of PEP was observed with the mechanistically distinct fructose-bisphosphate aldolase class I (Baldwin & Perham, 1978). Overall, we could validate the regulatory function of five out of eight newly observed enzyme–metabolite interactions.

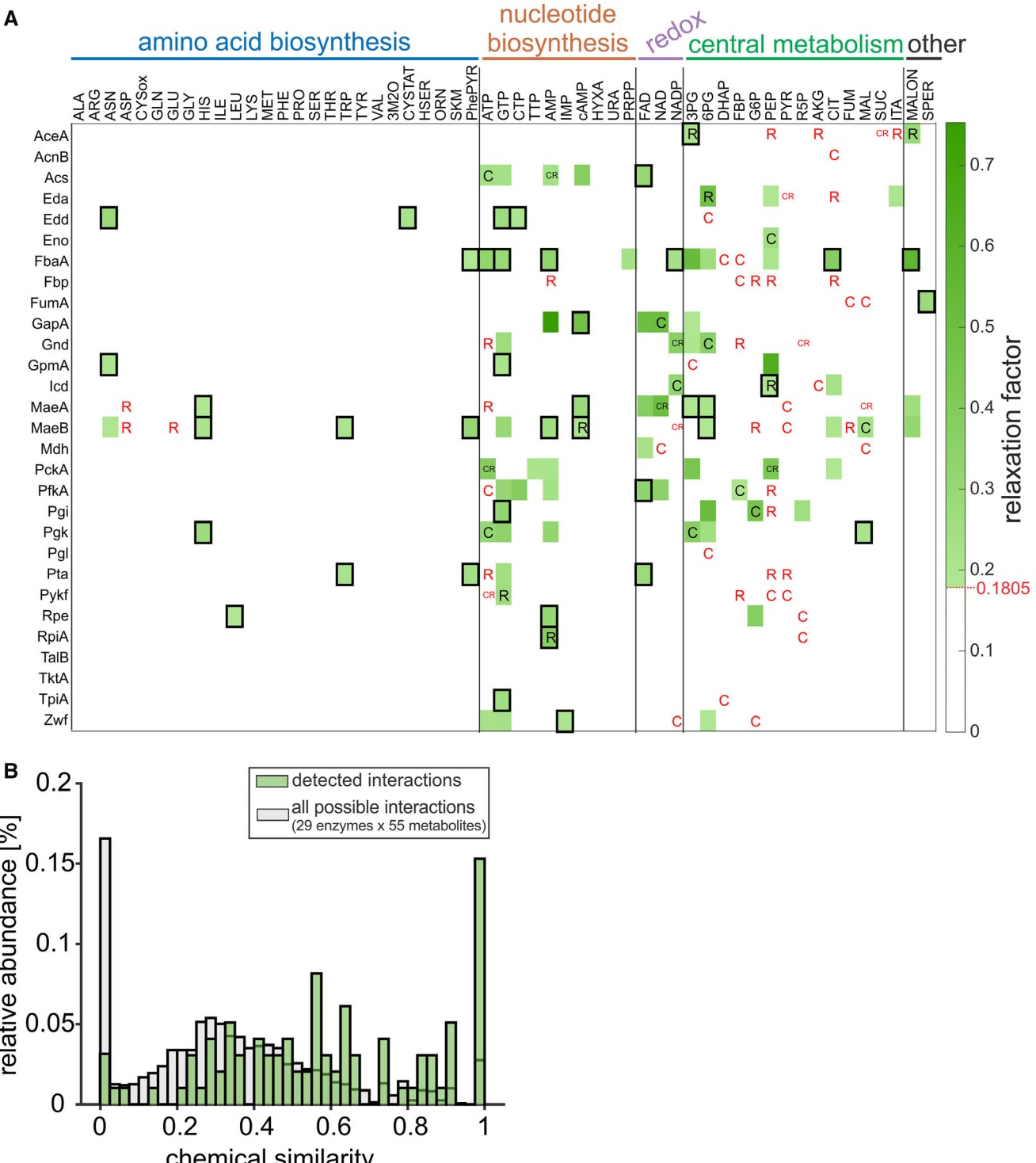

**Figure 4. A map of the enzyme–metabolite interactions in *Escherichia coli* central metabolism.**

A  Interactions between 29 central enzymes (in rows) and 55 metabolites (in columns), grouped according to metabolic pathways (*n* = 2, measurement replicates from the same sample). The relaxation factor of every interaction is indicated in green. Previously reported catalytic and regulatory interactions are denoted with "C" and "R", respectively; black and red letters indicate interactions that were detected and not detected, respectively. Black rectangles indicate potential allosteric interactions (maximum chemical similarity between the interactor and substrates/products of the target is lower than 0.5).

B  Histogram showing the relative occurrences of maximum chemical similarity scores in the protein–metabolite interaction map. Gray bars indicate the distribution of scores considering all possible enzyme–metabolite pairs, and green bars indicate the distribution scores for interactions with ΔRF > 0.1805 (false-positive rate < 5%).

Data information: Abbreviations of proteins and metabolites are explained in Dataset EV1.

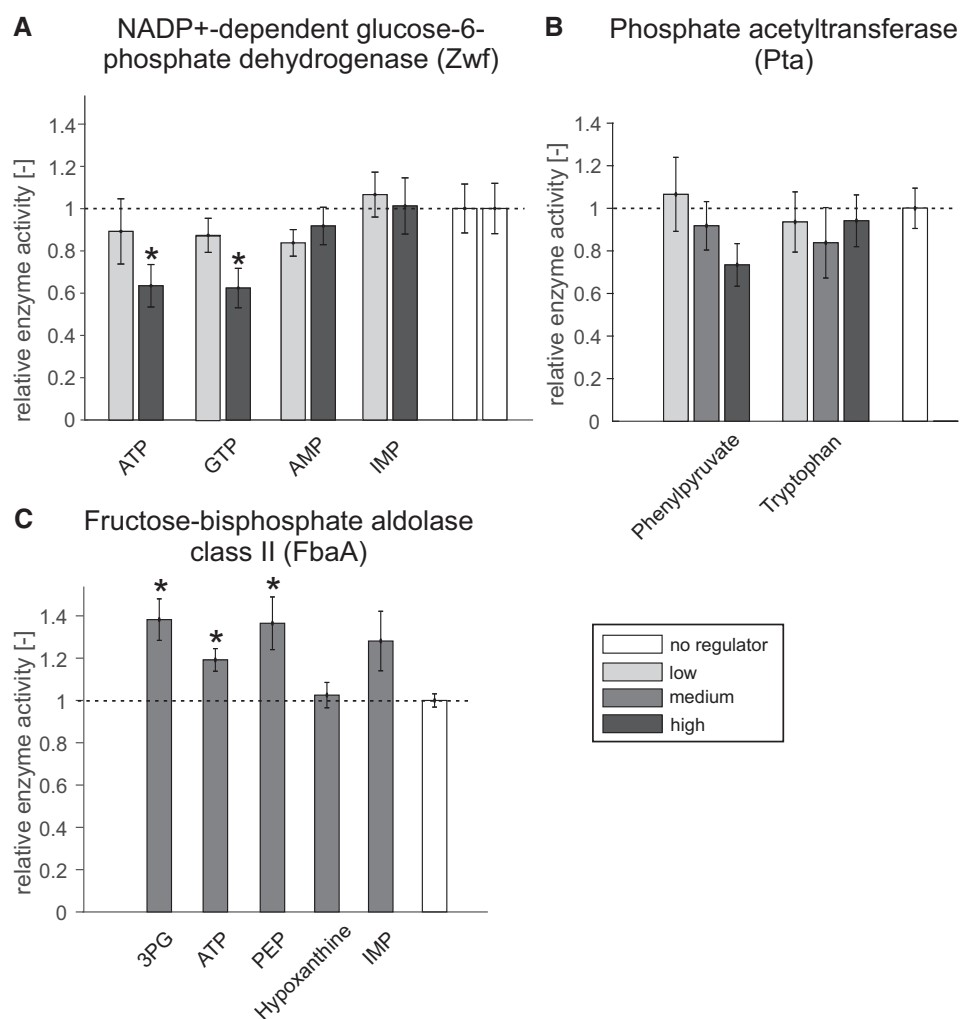

**Figure 5. *In vitro* enzyme assays.**

A   Relative activity of NADP⁺-dependent glucose-6-phosphate dehydrogenase in the presence of potential regulators; error bars represent the s.e.m. All regulators were tested at two concentrations in four distinct replicates ($n = 4$, ATP: 5 mM, 18 mM; GTP: 2 mM, 10 mM; IMP and AMP: 1 mM, 5 mM), and the asterisk denotes significant inhibition (18 mM ATP: *P*-value = 0.029; 10 mM GTP: *P*-value = 0.025, one-tailed *t*-test).

B   Relative activity of phosphate acetyltransferase in the presence of phenylpyruvate and tryptophan; error bars represent the s.e.m. All regulators were tested at three concentrations in three distinct replicates ($n = 3$, phenylpyruvate: 0.1, 1, 5 mM; ʟ-tryptophan: 0.1, 1, 4.4 mM).

C   Relative activity of fructose-bisphosphate aldolase class II in the presence of 5 mM of potential regulators, except hypoxanthine, which was tested at 3.22 mM. All regulators were tested in three distinct replicates ($n = 3$), and error bars represent the s.e.m. The asterisk denotes significant activation (5 mM 3PG: *P*-value = 0.025; 5 mM ATP: *P*-value = 0.024; 5 mM PEP: *P*-value = 0.046, one-tailed *t*-test).

Data information: The raw data for all *in vitro* enzyme assays can be found in Dataset EV3.

## Discussion

To probe the depth of our present knowledge on protein–metabolite interactions, we systematically mapped protein–(polar) metabolite interactions in the arguably best-characterized molecular network: *E. coli* central metabolism (Keseler *et al*, 2013; Placzek *et al*, 2017). For this purpose, we developed a higher throughput version of a ligand-detected NMR assay that was recently showcased to provide direct readout of binding events at high sensitivity for weak interactions (Nikolaev *et al*, 2016). Even with very conservative cutoffs at a 5% false-positive rate of the known interactions, the metabolite-binding profiles of 29 central enzymes with 55 metabolites identified 76 novel interactions. Detected interactions were spread across most enzymes, rather than focusing on few regulatory hubs. Although our NMR assays do not provide direct functionality evidence, we estimate that over 60% of the newly detected binding events are functional, based on the recovery of positive controls from protein–metabolite interaction databases and functional *in vitro* assays. By constructing the first near-comprehensive map of protein–(polar) metabolite interactions in a defined biological subnetwork, we could essentially double the number of known interactions. Given that central metabolism was already heavily

investigated, our results suggest that the presently available information about protein–metabolite interactions may only be the tip of the iceberg.

Most striking was the large number of interactions with purine nucleotides, most prominently with GTP. While ATP was known to bind to a variety of proteins in several organisms (Reinhard *et al*, 2015; Geer & Fitzgerald, 2016; Piazza *et al*, 2018), promiscuous binding of GTP was reported only recently (Piazza *et al*, 2018). The frequent binding of purine nucleotides does not appear to be explained by propensity of specific proteins for RNA binding, since we found no correlation with RNA binding enzymes from two recently published *E. coli* studies (Queiroz *et al*, 2019; Shchepachev *et al*, 2019). The recent discovery of purine nucleotide function as biological hydrotropes (Patel *et al*, 2017) could offer an explanation; however, effects on protein aggregation were only observed at much higher concentrations than used in our assay (> 5 mM compared to 0.2 mM). In general, purine nucleotide binding might induce coupling of enzyme activity to the availability of energy cofactors, although that does not explain the prominence of GTP over ATP. Alternatively, the large number of purine compared to pyrimidine interactions may simply reflect a general preference for purine-based regulators, as is also seen for second messengers such as cAMP and cGMP (Seifert, 2015; Nelson & Breaker, 2017). In contrast to nucleotides, amino acids interacted with few central enzymes. Thus, central metabolism appears to be regulated primarily through central metabolites and cofactors. In general, regulatory interactions are more frequent in glycolysis and pentose phosphate pathway, while we found much fewer interactions for TCA cycle enzymes. This observation is consistent with previous reports about primarily transcriptional and comparatively less metabolite regulation of the TCA cycle (Gerosa *et al*, 2015; Reznik *et al*, 2017). Remarkably, the two tested malic enzymes (MaeA and MaeB), which catalyze the decarboxylation of the TCA cycle intermediate malate to pyruvate, exhibit many new metabolite interactions. Possibly, this reflects the need for a tight regulation of reactions branching from the TCA cycle as suggested before (Reznik *et al*, 2017).

Enzyme activity modulation through metabolite binding may be achieved through competitive or allosteric interactions, i.e., binding at the active site or elsewhere on the protein, respectively. About 60% of the detected binders were of high chemical similarity (> 0.5) to the native reactants of their protein targets, suggesting competitive binding at the active site. This conclusion concurs with the observation that most known metabolite regulators are chemically similar to either the substrates or products of their enzyme target (Alam *et al*, 2017). The remaining 40 interactions with low chemical similarity scores (≤ 0.5) are hence suspected to allosterically bind at some distance from the active site. Using *in vitro* enzyme assays, we confirmed the regulatory role of five out of eight tested interactions, in particular showing that the branch point enzyme Zwf is inhibited by the frequently interacting metabolites ATP and GTP. In summary, of the five interactions showing significant effects on enzyme activity, two metabolites show the effect within their physiological concentration range (Zwf–GTP and FbaA–ATP), and three more show the effect at concentrations one-to sevenfold higher than the anticipated physiological steady state concentration range of the metabolite (Zwf–ATP, FbaA–3PG, and FbaA–PEP; Park *et al*, 2016; Kochanowski *et al*, 2017; Appendix Fig S10).

The previously published ligand-detected NMR experiments identified 15 interactions for enolase and 6-phosphofructokinase I (Nikolaev *et al*, 2016), while we report only four here, two of which were detected in both datasets (Eno–PEP and PfkA–AMP). The lower number of detected interactions in the current dataset is likely due to a combination of (i) a new assay buffer with physiological salt concentrations, to minimize unspecific binding; and (ii) more stringent analysis, including fully automated peak picking and exclusion of metabolite signals that are unstable over time. Our experiments included 27 proteins and 18 metabolites that were also covered in a recent proteomic study based on limited proteolysis coupled to mass spectrometry (LiP-MS; Piazza *et al*, 2018). The overlap between both studies was 11 interactions, six of which were not reported before, similarly to what can be expected from other large-scale studies (Diether & Sauer, 2017). Additionally, LiP-MS and our NMR analysis yielded 56 and 37 non-overlapping interactions on the same set of proteins and metabolites, respectively. These differences could result from NMR reporting on direct interactions under *in vitro* conditions, while LiP-MS senses both direct and indirect effects under native cellular extract conditions. Furthermore, while T1rho NMR is most sensitive to interactions with μM dissociation constants (Nikolaev *et al*, 2016), LiP-MS exhibits a broader sensitivity range (Piazza *et al*, 2018). Despite the many non-overlapping interactions, the amount of interactions detected for the same metabolites was comparable (correlation with $R^2 = 0.2857$; Appendix Fig S11). Notably, the true-positive rate in NMR experiments was twice as high as in LiP-MS (30% vs. 15.6%) at a comparable false-positive rate (5% vs. 5.5%) (Piazza *et al*, 2018). Our ligand-detected NMR method is therefore complementary to recent mass spectrometry-based proteomics approaches that are also unbiased with regard to specific metabolite and protein classes (Lomenick *et al*, 2009; Savitski *et al*, 2014; Piazza *et al*, 2018). The *in vitro* NMR assays are less susceptible to indirect effects, work also for low-abundance proteins and proteins with unfavorable proteolytic patterns, permit higher throughput in the number of tested metabolites, and could potentially be applied to metabolite-interacting RNA targets. MS-proteomics on the other hand permits working in native cellular extracts and provides higher throughput in the number of tested proteins, and in the case of limited proteolysis, provides also information on potential allosteric binding sites.

Systematic mapping of regulatory protein–metabolite interactions remains a challenge because of the weak energy of such interactions and only indirect detection offered by most existing methods. In this first large-scale application of the previously developed ligand-detected NMR approach (Nikolaev *et al*, 2016), we demonstrate the potential for exhaustive mapping of interactions in a pre-selected space of proteins and metabolites, providing direct readout of binding events and sensitivity to weak interactions without restrictions to specific protein or metabolite classes. With the here-established fully automated data analysis pipeline, including NMR spectra conversion, peak detection, and quantification of interactions, ligand-detected NMR can become an invaluable tool for the discovery of functional protein–metabolite interactions in different biological networks, including transcription factor networks and signaling pathways.

# Materials and Methods

## Reagents and Tools table

| Reagent/Resource | Reference or source | Identifier or catalog number |
| --- | --- | --- |
| **Experimental models** | | |
| ASKA library (*E. coli*) | Kitagawa *et al* (2005) | N/A |
| **Chemicals, enzymes and other reagents** | | |
| Yeast extract (for LB medium) | BD Bioscience | 288620 |
| Tryptone (for LB medium) | LLG Labware | 6271005 |
| Chloramphenicol | Merck (Sigma-Aldrich) | C0378 |
| Isopropyl β-D-1-thiogalactopyranoside | Merck (Sigma-Aldrich) | I5502 |
| B-PER™ reagent (in phosphate buffer) | Thermo Scientific | 78266 |
| Lysozyme from chicken egg white | Merck (Sigma-Aldrich) | 62970-5G-F |
| DNase I | PanReac AppliChem | A3778, 0100 |
| Phenylmethanesulfonyl fluoride | Merck (Sigma-Aldrich) | 93482 |
| His GraviTrap TALON columns | GE Healthcare | 29000594 |
| ZEBA™ Spin Desalting Columns, 7K MWCO, 10 ml | Thermo Scientific | 89893 |
| Imidazole | Merck (Sigma-Aldrich) | I3386 |
| di-Potassium hydrogen phosphate | Merck (Sigma-Aldrich) | 1.05104 |
| Potassium dihydrogen phosphate | Merck (Sigma-Aldrich) | 1.04873 |
| Trizma base | Merck (Sigma-Aldrich) | T1503 |
| NaCl | Merck (Sigma-Aldrich) | 1.06404 |
| KCl | Merck (Sigma-Aldrich) | 60130 |
| $MgCl_2$ | Merck (Sigma-Aldrich) | 63072 |
| $D_2O$ | ARMAR Chemicals | 014400 |
| 4,4-dimethyl-4-silapentane-1-sulfonic acid | Merck (Sigma-Aldrich) | 178837 |
| Methanol | Merck (Sigma-Aldrich) | 1.06009 |
| Isopropanol | Merck (Sigma-Aldrich) | 34863 |
| Water | Merck (Sigma-Aldrich) | 34877 |
| $NH_4F$ | Merck (Sigma-Aldrich) | 338869 |
| Hexakis(1H, 1H, 3H-tetrafluoropropoxy)phosphazine | Agilent | G1969-85001 |
| Homotaurine | Merck (Sigma-Aldrich) | A76109 |
| D-glucopyranose-6-phosphate | Merck (Sigma-Aldrich) | G7879 |
| Glyceraldehyde 3-phosphate | Merck (Sigma-Aldrich) | G5251 |
| Acetyl phosphate | Merck (Sigma-Aldrich) | 01409 |
| Coenzyme A | Merck (Sigma-Aldrich) | C4780 |
| Acetyl-coenzyme A | Merck (Sigma-Aldrich) | A2056 |
| ʟ-Alanine | Merck (Sigma-Aldrich) | 05130 |
| ʟ-Arginine | Merck (Sigma-Aldrich) | 11010 |
| ʟ-Asparagine | Merck (Sigma-Aldrich) | A8381 |
| ʟ-Aspartate | Merck (Sigma-Aldrich) | 11189 |
| ʟ-Cysteine | Merck (Sigma-Aldrich) | C7352 |
| ʟ-Glutamine | Merck (Sigma-Aldrich) | 49419 |
| ʟ-Glutamate | Merck (Sigma-Aldrich) | 49621 |
| Glycine | Merck (Sigma-Aldrich) | G5417 |
| ʟ-Histidine | Merck (Sigma-Aldrich) | 53320 |
| ʟ-Isoleucine | Merck (Sigma-Aldrich) | I2752 |

**Reagents and Tools table** (continued)

| Reagent/Resource | Reference or source | Identifier or catalog number |
|---|---|---|
| L-Leucine | Merck (Sigma-Aldrich) | 61819 |
| L-Lysine | Merck (Sigma-Aldrich) | L5501 |
| L-Methionine | Merck (Sigma-Aldrich) | M9625 |
| L-Phenylalanine | Merck (Sigma-Aldrich) | P2126 |
| L-Proline | Merck (Sigma-Aldrich) | P0380 |
| L-Serine | Merck (Sigma-Aldrich) | 84960 |
| L-Threonine | Merck (Sigma-Aldrich) | T8625 |
| L-Tryptophan | Merck (Sigma-Aldrich) | 93660 |
| L-Tyrosine | Merck (Sigma-Aldrich) | 93830 |
| L-Valine | Merck (Sigma-Aldrich) | V0500 |
| Phenylpyruvate | Merck (Sigma-Aldrich) | 286958 |
| Shikimate | Merck (Sigma-Aldrich) | S5375 |
| L-Homoserine | Merck (Sigma-Aldrich) | H6515 |
| 3-Methyl-2-oxobutanoate | Merck (Sigma-Aldrich) | 198994 |
| L-Ornithine | Merck (Sigma-Aldrich) | O8305 |
| L-Cystathionine | Merck (Sigma-Aldrich) | C7505 |
| 2-Oxoglutarate | Merck (Sigma-Aldrich) | 75890 |
| Citrate | Merck (Sigma-Aldrich) | 1.00244 |
| L-Malate | Merck (Sigma-Aldrich) | M1125 |
| Succinate | Merck (Sigma-Aldrich) | S2378 |
| Fumarate | Merck (Sigma-Aldrich) | F1506 |
| Itaconate | Merck (Sigma-Aldrich) | 129204 |
| Nicotinamide adenine dinucleotide | Merck (Sigma-Aldrich) | 43410 |
| Nicotinamide adenine dinucleotide phosphate | Merck (Sigma-Aldrich) | N5755 |
| Flavin adenine dinucleotide | Merck (Sigma-Aldrich) | F6625 |
| D-Glucose-6-phosphate | Merck (Sigma-Aldrich) | G7250 |
| D-Fructose 1,6-bisphosphate | Merck (Sigma-Aldrich) | F6803 |
| Dihydroxyacetone phosphate | Merck (Sigma-Aldrich) | 51269 |
| D-(-)-3-Phosphoglycerate | Merck (Sigma-Aldrich) | P8877 |
| Phosphoenolpyruvate | Merck (Sigma-Aldrich) | 860077 |
| Pyruvate | Merck (Sigma-Aldrich) | P2256 |
| 6-Phosphogluconate | Merck (Sigma-Aldrich) | P7877 |
| D-Ribulose-5-phosphate | Merck (Sigma-Aldrich) | 83899 |
| Inosine 5′-monophosphate | Merck (Sigma-Aldrich) | I4500 |
| Hypoxanthine | Merck (Sigma-Aldrich) | H9377 |
| 5-Phospho-D-ribose 1-diphosphate | Merck (Sigma-Aldrich) | P8296 |
| Adenosine 5′-monophosphate | Merck (Sigma-Aldrich) | 01930 |
| Adenosine 5′-triphosphate | Merck (Sigma-Aldrich) | A26209 |
| Adenosine 3′,5′-cyclic monophosphate | Merck (Sigma-Aldrich) | A6885 |
| Cytidine 5′-triphosphate | Merck (Sigma-Aldrich) | C1506 |
| Guanosine 5′-triphosphate | Merck (Sigma-Aldrich) | G8877 |
| Thymidine 5′-triphosphate | Merck (Sigma-Aldrich) | T0251 |
| Uracil | Merck (Sigma-Aldrich) | 94220 |
| Spermidine | Merck (Sigma-Aldrich) | S2626 |
| Malonate | Merck (Sigma-Aldrich) | M1875 |

**Reagents and Tools table** (continued)

| Reagent/Resource | Reference or source | Identifier or catalog number |
|---|---|---|
| **Software** | | |
| Matlab R2018b | The MathWorks, Inc. | N/A |
| Python 2.5.3 (TopSpin-Jython implementation) | https://www.python.org/ | N/A |
| TopSpin 3.2 | Bruker | N/A |
| NMRmix | Stark *et al* (2016) | N/A |
| Simcomp2 | Hattori *et al* (2010); https://www.genome.jp/tools/simcomp2/ | N/A |
| **Other** | | |
| 5-mm TA tubes | ARMAR Chemicals | N/A |
| Avance III-HD 600 MHz NMR spectrometer, CPTCI 1H/19F-13C/15N-2H probe | Bruker | N/A |
| Tecan Infinite M200 | Tecan | N/A |
| 6520 series iFunnel quadrupole time-of-flight mass spectrometer | Agilent | N/A |
| MPS2 autosampler | GERSTEL | N/A |

## Methods and Protocols

### Protein expression and purification

LB shake flask cultures (200–600 ml, 5 g/l yeast extract, 10 g/l tryptone, 10 g/l NaCl) supplemented with 100 µg/ml chloramphenicol were inoculated in a 1:100 ratio with LB precultures of the overexpression strains (Kitagawa *et al*, 2005), and expression was induced with 0.1 mM isopropyl β-D-1-thiogalactopyranoside. Cultures were incubated for 16 h at 37°C while being shaken (300 rpm). Cells were harvested by centrifugation (5,000 *g* and 4°C for 15 min) and flash-frozen in liquid nitrogen. For cell lysis, cells were resuspended in lysis buffer [B-PER reagent in phosphate buffer (Thermo Scientific) supplemented with 500 mM NaCl, 20 mM imidazole, 4 mM phenylmethanesulfonyl fluoride, 1 mM MgCl$_2$, 2 mg/ml lysozyme, and 0.2 mg/ml DNase I (PanReac AppliChem), volume: 10% of culture volume]. The suspension was shaken at room temperature for 10 min, and cell extracts were separated from cell debris by centrifugation (20,000 *g* and 4°C for 30 min). His-tagged proteins were purified from cell extracts using His GraviTrap TALON gravity flow columns (GE Healthcare), and the elution buffer was replaced by the respective assay buffer [20 mM potassium phosphate buffer (pH 7.5), 100 mM KCl, 10 mM NaCl, and 5 mM MgCl$_2$] using ZEBA™ spin desalting columns with 7 kDa cutoff (Thermo Scientific). The purity of all tested proteins was higher than 90%, as assessed by sodium dodecyl sulfate–polyacrylamide gel electrophoresis (SDS–PAGE; Appendix Fig S12), in agreement with the study in which these overexpression strains had first been described. The purified proteins were flash-frozen in liquid nitrogen and stored at −80°C until further usage.

### NMR measurements and analysis

#### Sample preparation (pure metabolite samples)

In order to obtain a reference peak list, NMR spectra of the pure metabolites were recorded in 425–500 µl volume containing 200 µM metabolite in assay buffer [20 mM potassium phosphate buffer (pH 7.5), 100 mM KCl, 10 mM NaCl, and 5 mM MgCl$_2$], 10% D$_2$O, and 25 µM 4,4-dimethyl-4-silapentane-1-sulfonic acid (DSS).

#### Sample preparation (proteins and metabolites)

Proteins stored at −80°C were thawed quickly and centrifuged (2 min, 20,000 *g*, room temperature) to remove aggregates. Protein concentrations were measured based on their specific extinction coefficients at 280 nm immediately before NMR sample preparation. In all experiments, the final protein and metabolite concentrations were 15 µM (monomer) and 200 µM in assay buffer, respectively. All samples were prepared in a total volume of 425–500 µl in 5-mm TA tubes (ARMAR Chemicals) and contained 10% D$_2$O and 25 µM DSS.

#### NMR measurements

NMR measurements were performed at 298K on a Bruker Avance III-HD 600 MHz spectrometer as described previously (Nikolaev *et al*, 2016). For proteins with final native mass below 40 kDa (Pgk, Pgl, Rpe, TpiA), T1rho experiments used 400 ms instead of 200 ms relaxation delay to compensate for lower relaxation rate enhancement than in larger proteins. For acquisition, a 512-scan T1rho experiment with long (200–400 ms) relaxation delay was sandwiched between two replicates of 256-scan T1rho experiments with short (10 ms) relaxation delays. In the final analysis, the two short-delay T1rho spectra were summed, thus averaging potential time-dependent differences in the peak intensities when compared to the long-delay T1rho spectra. Each acquisition included a 1D-$^1$H spin-echo experiment (Sklenář & Bax, 1987), which provides maximal sensitivity for the signals of proteins with large molecular weight, thus allowing the control of protein stability between the samples. To automate the data acquisition, a Python-based library for sample changing, basic experiment setup, and acquisition logging was developed (github.com/systemsnmr/metabolite-interactions), making use of the Bruker "NMR Case" accessory.

#### NMR processing and data analysis

Spectra were processed in TopSpin 3.2 (Bruker) using custom-built Python routine to process, calibrate, and calculate the difference spectra (github.com/systemsnmr/metabolite-interactions). Calibration of spectra to the DSS reference signal was critical to minimize

subtraction artifacts in the final difference spectra. As a measure of experimental reproducibility and a quality filter for metabolite stability, the two short-delay T1rho replicate spectra for each sample were compared. Metabolite signals showing more than 5% difference between these two spectra were considered unstable in the presence of the given protein and were excluded from the final analysis. Additionally, metabolite peaks that appeared to gain intensity in the presence of the protein were excluded. These were identified as the peaks which showed a marked negative intensity (smaller than −0.05) after subtracting the intensities of the combined short-delay spectra of the protein–metabolite mixtures (T1rho10ms_PM), and free protein (T1rho10ms_P), from the free metabolite reference spectra (T1rho10ms_M): [T1rho10ms_M − (T1rho10ms_PM − T1rho10ms_P)] < −0.05].

### Peak detection and quantification

Identification, signal-to-noise ratio (S/N) quantification, assignment, and disambiguation of interaction hits were performed using custom-built Matlab scripts (github.com/systemsnmr/metabolite-interactions). For every peak of every compound, a peak window was determined. After the optimal metabolite mixes were identified with the NMRmix software (Stark *et al*, 2016), a list with peak windows for all metabolites was compiled for every metabolite mix, excluding the regions with overlapping peaks. During further analysis of NMR spectra, maximum peak intensities in a given spectrum were identified by scanning the pre-defined peak window for the maximum value. The S/N for every peak was calculated from the T1rho NMR spectrum of the protein–metabolite mixtures, after subtracting pure protein and pure metabolite signals.

### Quantification of interactions with the relaxation factor

Interactions between enzymes and metabolites were quantified by computing the relaxation factor (Gossert & Jahnke, 2016; Equation 1). In brief, this metric measures the difference in relaxation of the metabolite alone and in the presence of the protein. In case there is no interaction, the relaxation of the metabolite is not affected by the presence of the protein yielding a low $\Delta$RF. If a protein–metabolite interaction is present, the relaxation factor decreases in the presence of the protein, yielding a value > 0 for $\Delta$RF. Specifically, maximum peak intensities in 10-ms-relaxation and 200-ms-relaxation (or 400 ms for proteins with molecular weight smaller than 40 kDa) NMR spectra were used to compute the relaxation factor.

$$\Delta RF = \left(\frac{M_{200ms}}{M_{10ms}}\right) - \left(\frac{(PM)_{200ms} - P_{200ms}}{(PM)_{10ms} - P_{10ms}}\right) \tag{1}$$

### Analysis of the recovery of known interactions

In order to analyze the predictive power of the relaxation factor, we investigated the recovery of known interactions for different relaxation factor cutoffs. We considered the total space of interactions as the combination of all tested enzymes (29) and all tested metabolites (55) yielding 1,595 possible interactions. Known interactions were taken from the EcoCyc database, whereby 72 unique catalytic and/or regulatory interactions were identified. Next, we calculated the false-positive rate (FPR) and true-positive

rate (TPR) for $\Delta$RF cutoffs in the interval [0, 0.5] according to the following formulas:

$$TPR_{\Delta RFC} = \frac{TP_{\Delta RFC}}{TP_{\Delta RFC} + FN_{\Delta RFC}}, \ \Delta RF \ cutoff \in [0, 0.5] \tag{2}$$

$$FPR_{\Delta RFC} = \frac{FP_{\Delta RFC}}{FP_{\Delta RFC} + TN_{\Delta RFC}}, \ \Delta RF \ cutoff \in [0, 0.5] \tag{3}$$

Here, true-positive (TP) and false-negative (FN) interactions refer to the number of known interactions with a $\Delta$RF above or below the cutoff $\Delta$RF cutoff, respectively. Accordingly, false-positive (FP) and true-negative (TN) interactions refer to the number of interactions not reported in EcoCyc with a $\Delta$RF above or below the cutoff $\Delta$RF cutoff, respectively. Note that this approach likely underestimates the total amount of true-positive interactions in the dataset, since only previously known interactions are considered true positive. Plotting the true-positive rate against the false-positive rate yields a receiver operating characteristic curve (ROC curve; Appendix Fig S5). For further investigation of the dataset, we selected a false-positive rate of 5%, corresponding to a $\Delta$RF cutoff of 0.1805.

### Metabolite mix assembly

The tool NMRmix was used to sort 59 metabolites (55 metabolites from this study, and CoA, acetyl-CoA, erythrose-4-phosphate, and 5-aminoimidazole-4-carboxamide ribonucleotide (AICAR)) into mixes with low spectral overlap (Stark *et al*, 2016). Peak lists generated from pure metabolite NMR spectra (see above) were used as input, and NMRmix was run using the following settings: *Optimizing*: Mixtures: 4, Max Mixture Size: 15, Cooling Rate: Linear, Start Temperature: 10,000, Final Temperature: 25, Max Steps: 500,000, Mix Rate: 2, Iterations: 10, Randomize Initial Mixtures: ON; *Refining*: Use Refinement: ON, Cooling Rate: Exponential, Start Temperature: 50, Final Temperature: 25, Max Steps: 1,000, Mix Rate: 2; and *Scoring*: Overlap Range: 0.025, Score Scaling: 100, Use Intensity Scoring: ON, and Autosave Results: ON.

NMR measurements of the metabolite mixes revealed degradation of CoA, acetyl-CoA, erythrose-4-phosphate, and AICAR, which were thus excluded from further experiments. To prevent potential enzymatic reactions, ATP (initially in mix 3) was exchanged with cAMP (initially in mix 1) and glucose-6-phosphate (initially in mix 4) was transferred to mix 2. The final mix compositions are given in Dataset EV1.

### Computation of chemical similarity

Chemical similarity of metabolite regulators and the substrates/products of the tested enzymes was calculated using the Web-based tool Simcomp2 (Hattori *et al*, 2010). The settings for "Global search" were applied, and the cutoff was lowered to 0.01.

### *In vitro* enzyme assays

#### Spectrophotometric assays for Zwf activity

Enzymatic *in vitro* assays for Zwf were performed by measuring the formation of NADPH photometrically at 340 nm. Assays were performed at 30°C in assay buffer [20 mM potassium phosphate buffer (pH 7.5), 100 mM KCl, 10 mM NaCl, and 5 mM MgCl$_2$] with

2 mM D-glucopyranose-6-phosphate and 1 mM NADP$^+$, and the competing metabolites at a concentration of 1–18 mM (ATP: 5 mM, 18 mM; GTP: 2 mM, 10 mM; IMP: 1 mM, 5 mM; AMP: 1 mM, 5 mM). Reactions were started by addition of the purified Zwf enzyme (final concentration in the assay of roughly 40 nM monomer), and the formation of NADPH was monitored photometrically at 240 nm every 10 s using a TECAN Infinite M200 spectrometer. Initial reaction velocities (within the first 120–150 s) were then determined by linear regression. Reaction velocities were normalized to the control without regulator, and the significance of inhibition was calculated using a one-sided Student *t*-test. Each experiment was performed in experimental quadruplicates. Raw data for the *in vitro* assay are provided in Dataset EV3.

### Mass spectrometry-based FbaA activity assays

FbaA activity assays were performed at 30°C in assay buffer [10 mM Tris–HCl pH 7.5, 1 mM MgCl$_2$] with 1 mM fructose 1,6-bisphosphate and the competing metabolites at a concentration of 3.22 or 5 mM (hypoxanthine: 3.22 mM; ATP: 5 mM; 3PG: 5 mM; PEP: 5 mM; IMP: 5 mM). Reactions were started by addition of the purified FbaA enzyme (final concentration in the assay of roughly 1 μM monomer). At the indicated time points, a 10-μl aliquot of the reaction solution was transferred to 40 μl methanol pre-cooled to −20°C to quench the reaction by inducing enzyme denaturation. Reactant concentrations were subsequently measured by flow-injection time-of-flight mass spectrometry (FIA TOF-MS). Negatively charged ions were putatively annotated based on accurate mass using 0.05 Da tolerance assuming simple deprotonation ([M-H]$^-$). Each experiment was performed in experimental triplicates. In order to distinguish between mass spectrometry artifacts and real concentration changes, product calibration curves were prepared. 0, 15, 30, and 60 μM of dihydroxyacetone phosphate and glyceraldehyde 3-phosphate were mixed with 1 mM fructose 1,6-bisphosphate in assay buffer [10 mM Tris–HCl pH 7.5, 1 mM MgCl$_2$], and the putative regulators were added at a concentration of 3.22 or 5 mM (hypoxanthine: 3.22 mM; ATP: 5 mM; 3PG: 5 mM; PEP: 5 mM; IMP: 5 mM). A 10-μl aliquot of each mix was transferred to 40 μl methanol pre-cooled to −20°C. Reactant concentrations were subsequently measured by FIA TOF-MS. Negatively charged ions were tentatively annotated based on accurate mass using 0.05 Da tolerance assuming simple deprotonation ([M-H] $^-$). Each calibration curve was performed in quadruplicates (Appendix Fig S13). Raw data for the *in vitro* assay and the calibration curves are provided in Dataset EV3.

### Mass spectrometry-based Pta activity assays

Pta activity assays were performed at 30°C in assay buffer [10 mM Tris–HCl pH 7.5, 1 mM MgCl$_2$] with 1 mM acetyl phosphate, 1 mM coenzyme A, and the competing metabolites at a concentration of 0.1–5 mM (phenylpyruvate: 0.1, 1, 5 mM; ʟ-tryptophan: 0.1, 1, 4.4 mM). Reactions were started by addition of the purified Pta enzyme (final concentration in the assay of roughly 200 nM monomer). At the indicated time points, a 10-μl aliquot of the reaction solution was transferred to 40 μl methanol pre-cooled to −20°C to quench the reaction by inducing enzyme denaturation. Reactant concentrations were subsequently measured by FIA TOF-MS. Negatively charged ions were putatively annotated based on accurate mass using 0.05 Da tolerance assuming simple deprotonation

([M-H]$^-$). Each experiment was performed in experimental triplicates. In order to distinguish between mass spectrometry artifacts and real concentration changes, product calibration curves were prepared. 0, 15, 30, and 60 μM of acetyl-coenzyme A were mixed 1 mM fructose 1,6-bisphosphate in assay buffer [10 mM Tris–HCl pH 7.5, 1 mM MgCl$_2$], and the putative regulators were added at a concentration of 0.1–5 mM (phenylpyruvate: 0.1, 1, 5 mM; ʟ-tryptophan: 0.1, 1, 4.4 mM). A 10-μl aliquot of each mix was transferred to 40 μl methanol pre-cooled to −20°C. Reactant concentrations were subsequently measured by FIA TOF-MS. Negatively charged ions were tentatively annotated based on accurate mass using 0.05 Da tolerance assuming simple deprotonation ([M-H]$^-$). Each calibration curve was performed in quadruplicates, except for 5 mM phenylpyruvate, where two curves had to be excluded due to empty injections (Appendix Fig S13). Raw data for the *in vitro* assay and the calibration curves are provided in Dataset EV3.

### Semi-quantitative measurement of metabolite concentrations using FIA TOF-MS

Samples were analyzed by direct flow double injection on an Agilent 6520 series iFunnel quadrupole time-of-flight mass spectrometer (Agilent, Santa Clara, CA, USA) coupled to a GERSTEL MPS2 autosampler (Fuhrer *et al*, 2011). Mass spectra were recorded in negative ionization mode within a mass/charge ratio range of 50–1,000. The mobile phase was 60:40 isopropanol:water (*v/v*) and 1 mM NH$_4$F at pH 9.0, supplemented with hexakis (1H, 1H, 3H-tetrafluoropropoxy) phosphazine and homotaurine for online mass correction. For every sample, two technical replicates were measured.

### Statistics

The Shapiro–Wilk test was used to confirm normality of the distribution of initial reaction velocities, and equality of variance was confirmed via Bartlett's test. Subsequently, initial reaction velocities were compared with *t*-tests. All statistical testing was performed using Matlab (The MathWorks, Inc.).

## Data availability

The datasets and computer code produced in this study are available in the following databases:
(i) Python code used for NMR experiment setup and spectra processing: GitHub (github.com/systemsnmr/metabolite-interactions).
(ii) Matlab code used for identification, signal-to-noise ratio (S/N) quantification, assignment, and disambiguation of interaction hits: GitHub (github.com/systemsnmr/metabolite-interactions).
(iii) Raw and processed NMR data: Zenodo (https://doi.org/10.5281/zenodo.3339911).

**Expanded View** for this article is available online.

### Acknowledgements

This work was supported by the SignalX project of the Swiss Initiative for Systems Biology (SystemsX.ch) and in part by the Promedica Stiftung, Chur (Grant 1300/M to Y.N.). We thank Karin Meier and Karin Ortmayr for helpful feedback and discussions.

*Maren Diether et al*

## Author contributions

MD, YN, and US designed the project and wrote the article. MD and YN designed the experiments, analyzed the data, and prepared the figures. MD performed most experiments and optimized the automated data analysis. YN automated NMR acquisition and data analysis and acquired NMR data. FHTA supervised the NMR aspects of the project and contributed to financial support.

## Conflict of interest

The authors declare that they have no conflict of interest.

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
