## [Review Process File · Molecular Systems Biology]

Systematic mapping of protein-metabolite interactions in central metabolism of *Escherichia coli*

Maren Diether, Yaroslav Nikolaev, Frédéric H.T. Allain and Uwe Sauer.

Review timeline:

Submission date:	17 th May 2019
Editorial Decision:	28 th June 2019
Revision received:	18 th July 2019
Accepted:	31 st July 2019

Editor: Maria Polychronidou

Transaction Report:

1st Editorial Decision

28th June 2019

Thank you again for submitting your work to Molecular Systems Biology. We have now heard back from the three referees who agreed to evaluate your study. As you will see below, the reviewers raise a series of concerns, which we would ask you to address in a revision.

Reviewers #1 and #2 are quite supportive and raise only a few issues that I think there is no need to repeat here. Reviewer #2, who would not be opposed to revealing their identity to you, has asked us to forward you a PDF with comments made directly on the manuscript text (attached below). These few comments on how to improve the study refer to clarifications and very minor text changes. Reviewer #3 is somewhat less supportive and raises concerns regarding the physiological and in vivo relevance of the reported interactions. We would ask you to clarify this issue in the revised manuscript. Please feel free to contact me in case you would like to discuss in further detail any of the issues raised.

REFEREE REPORT

Reviewer #1:

The authors present their application of ligand detected NMR to the enzymes of central carbon metabolism in *E.coli*. Using this established approach, they identify candidate allosteric regulators of several of these enzymes and validate a few interactions in vitro.

The addition of two pieces of information would help to place these results in biological context.

First, the authors describe a successive data acquisition used to rule out "unstable" compounds. Are some of these in fact the result of catalysis? If so, this bears further discussion in the text and

perhaps in the supplementary data as well, as it provides potential guidance about the substrate specificity. (Of course, contaminating activities may be an issue, but this can be discussed.)

In addition, the potential regulators can be put into better context by also providing the literature values for metabolite concentration. The concentrations tested here are in typical physiological range for some compounds, but well above for others. While an effect at high concentration may well be relevant in a specific physiological context, including the typical concentrations of these regulators and potential regulators will help the audience put them in context for their relevance to typical conditions.

Reviewer #2:

Regulation of enzymes via metabolites is a critical layer of cell regulation. Yet, due to limited technical capabilities it is a notoriously limited in its characterisation.

With that situation as background, new techniques are urgently needed, and the current paper is the most impressive exploration of such an approach I saw. I view this paper as highly deserving of publication. I wrote specific comments on the manuscript which I was hoping to communicate to the authors.

I do not have an objection that my identity will be revealed via the commented manuscript

Reviewer #3:

The authors studied metabolite protein interactions in the e.coli central metabolism. Therefore 29 proteins were expressed and purified and tested with 4 mixtures containing in total 55 metabolites.

The protein-metabolite interactions were analyzed using NMR based methods.

The authors report that they found 98 interactions with 76 being novel.

However, only 30% of the known interactions could be verified and only 5 out of 8 novel interactions that the authors scored as allosteric were shown regulatory.

Furthermore the functionality of these interactions was found at extreme high concentrations of the interacting metabolites (e.g. 10mM).

Do the authors see the opportunity that such high concentrations can be achieved in vivo?

Why could the known interactions not be retrieved?

In this stage the study is very preliminary and does not lead to novel insights or regulatory principles, yet. The regulatory interactions seem at extreme concentrations, thus the relevance upon 'normal' metabolic regulation may be questioned.

If one would calculate the success of the method to measure the known interactions and the frequency that identified allosteric binders influence the activity of the target enzyme at extreme conditions the publication is not mature at this stage.

1st Revision - authors' response

18th July 2019

Reviewer #1:

The authors present their application of ligand detected NMR to the enzymes of central carbon metabolism in E.coli. Using this established approach, they identify candidate allosteric regulators of several of these enzymes and validate a few interactions in vitro.

The addition of two pieces of information would help to place these results in biological context.

First, the authors describe a successive data acquisition used to rule out "unstable" compounds. Are some of these in fact the result of catalysis? If so, this bears further discussion in the text and perhaps in the supplementary data as well, as it provides potential guidance about the substrate specificity. (Of course, contaminating activities may be an issue, but this can be discussed.)

It is indeed conceivable that some of the compounds ruled out as 'unstable' were subject to catalysis by proteins. However, because protein-metabolite mixtures were incubated at room temperature for ~3-8 h before the actual NMR measurement, we assume that any significant catalytic conversions should already take place before the NMR measurements and thus, would not be observable during successive experimental acquisitions. Differentiating catalytic activity from other sources of compound instability is thus not easily possible in our current set up. Nevertheless, the data on such potential "catalytic" events is preserved in our datasets. We added a statement addressing this point in the first 'Results' paragraph:

"Among other sources of instability, metabolite degradation could be the result of enzymatic conversion, although this is not likely to be a major confounding factor given that the protein metabolite mixture was incubated for several hours prior to NMR recording. However, differentiating the various sources of metabolite instability is not feasible given our current setup. "

In addition, the potential regulators can be put into better context by also providing the literature values for metabolite concentration. The concentrations tested here are in typical physiological range for some compounds, but well above for others. While an effect at high concentration may well be relevant in a specific physiological context, including the typical concentrations of these regulators and potential regulators will help the audience put them in context for their relevance to typical conditions.

To help readers to put our results into the context of typical cellular conditions we added an Appendix figure (S10) comparing physiological metabolite concentrations (from Park 2016 and Kochanowski 2017) with the concentrations used in our *in vitro* enzyme assays. Note that the anticipated "physiological concentrations" were sampled from only a few experiments with narrow set of conditions each, and thus likely represent only a part of the full physiological concentration range. In summary, of the five interactions showing confident effect on enzyme activity, two metabolites show the effect within their physiological concentration range (Zwf-GTP, FbaA-ATP), and three more show the effect at concentration 1- to 7-fold higher than the anticipated physiological concentration range of the metabolite (Zwf-ATP, FbaA-3PG, FbaA-PEP). We included a statement on physiological concentration ranges in the third paragraph in the 'Discussion' section.

In summary, of the five interactions showing significant effects on enzyme activity, two metabolites show the effect within their physiological concentration range (Zwf-GTP, FbaA-ATP), and three more show the effect at concentrations 1- to 7-fold higher than the anticipated physiological steady state concentration range of the metabolite (Zwf-ATP, FbaA-3PG, FbaA-PEP) (Park et al, 2016; Kochanowski et al, 2017) (Appendix Fig. S10).

More generally, the NMR experiments used in our study detect interactions with dissociation constants (K_D s) in the μM -to-mM range (mentioned in the last paragraph of section 'Results, Ligand-detected T1rho NMR assay for a biological subnetwork').

Reviewer #2:

Regulation of enzymes via metabolites is a critical layer of cell regulation. Yet, due to limited technical capabilities it is a notoriously limited in its characterisation.

With that situation as background, new techniques are urgently needed, and the current paper is the most impressive exploration us such an approach I saw. I view this paper as highly deserving of publication. I wrote specific comments on the manuscript which I was hoping to communicate to the authors.

I do not have an objection that my identity will be revealed via the commented manuscript

Comments of reviewer 2 (extracted from pdf annotation)

Comments regarding the introduction

Sentence: One challenge is the generally low affinity of protein-metabolite interactions

Comment: It would be useful in my opinion if an order of magnitude was given. Say mM.

We adapted the sentence as follows: *One challenge is the generally low affinity (mM range) of protein-metabolite interactions (Reznik et al, 2017), and their fleeting nature.*

Sentence: At present about 100 regulatory and 130 catalytic interactions involving ...

Comment: Would be useful to define regulatory versus catalytic interactions, maybe via an example.

We adapted the sentence as follows: *At present about 100 regulatory (metabolite changes enzyme activity) and 130 catalytic (metabolite is substrate or product) interactions involving the 35 major isoenzymes of central metabolism are reported in the EcoCyc database (Keseler et al, 2017).*

Sentence: ... by choosing cutoffs that recovered 30% of all known interactions at a false-positive rate of 5%.

Comment: This sounds to be low, please motivate the choice

We rephrased the sentence to increase clarity: *Here, we focused our analysis only on high confidence NMR interactions by choosing a false-positive rate cutoff of 5%, which yielded a dataset encompassing 30% of the 72 known interactions.*

Sentence: At this cutoff, we detected 98 interactions between all tested enzymes and metabolites, including 76 interactions that had not been reported previously,

Comment: how many of the known interactions were recovered? 30% of the 100+130 mentioned above?

We included the following statement: *"We systematically generated ligand-detected NMR interaction profiles of 29 purified enzymes from E. coli central metabolism with 55 selected metabolites, between which 72 interactions were already known."*

Comments regarding the results section: Ligand-detected T1rho NMR assay for a biological subnetwork

Sentence: Additionally, we selected metabolites from branch points of metabolic pathways, based on the assumption that these are more likely to exert regulatory roles.

Comment: is there a ref to this?

Previous studies have shown that metabolic regulation is more likely to occur at branch point enzymes. However, preferential regulatory roles for branch point metabolites have not been demonstrated yet. We removed the corresponding sentence to avoid confusion.

Sentence: To detect protein-metabolite interactions, purified proteins were mixed with a subset of metabolites and NMR spectra were recorded.

Comment: Please explain the method in a few more sentences to people who have not read the previous article.

We added more sentences to clarify the method: *A single one-dimensional (1D) NMR spectrum can resolve few dozens of individual metabolite signals. Due to differences in the NMR properties of small and large molecules, metabolite signals broaden (exhibit reduced intensity) upon protein binding. We exploit this change in signal intensity to detect metabolite-protein interactions.*

Sentence: 1D1H T1rho

Comment: Is this defined anywhere?

We specified the definition of 1D1H in the main text (*'one-dimensional hydrogen-detected'*). The physical explanation of the term "T1rho relaxation" is detailed in the reference citation provided at the end of the corresponding sentence.

Sentence: Water-LOGSY,

Comment: Is this defined anywhere?

We added an explanation for the abbreviation: *water-ligand observed via gradient spectroscopy (Water-LOGSY)*. More detailed explanation is included in our earlier, pilot study, which is cited in the sentence where the term is used.

Comments regarding the results section: Systematic map of protein-metabolite interactions in E. coli central metabolism

Sentence: we calculated the false-positive and ...

Comment: Please clarify - How can one tell if something is a false-positive if not all interactions are known yet? That is, how do you know something is false.

For this calculation, we assume that all previously reported interactions (from EcoCyc) are true positives, whereas all other interactions are true negatives. With this assumption, we likely underestimate the total number of true positive interactions, suggesting that the actual True Positive Rate at any given False Positive Rate cutoff is better than suggested by our estimates in the ROC curve analysis (ROC

curve, Appendix Fig S5). We added a reference to the corresponding methods section, where this is explained in more detail.

Sentence: we detected 98 distinct protein-metabolite interactions ...

Sentence: We recovered 22 of the 72 previously reported interactions ...

Comment: maybe start with this after stating the 98.

We moved this sentence further up, to appear after the statement on '98 interactions ...'

Sentence: Piazza 2018, 15.6% recovery

Comment: Better to use 2 significant digits not 3.

Implemented.

Comments regarding the results section: Chemical similarities distinguish between potential allosteric and competitive interactions

Sentence: using Simcomp2

Comment: Maybe write a sentence on what this is and how it works for readers who do not know about it.

We added a sentence to clarify the method: *Simcomp2 identifies the maximal common substructure of two chemical structures using a graph-based method.*

Sentence: Nevertheless, 40% of the NMR-detected interactors have a low chemical similarity (< 0.5) to substrates/products of ...

Comment: It would be interesting to see a histogram of the distribution, maybe in an SI figure

We added a reference to Figure 4B after this sentence, to point readers towards the histogram.

Reviewer #3:

The authors studied metabolite protein interactions in the e.coli central metabolism. Therefore 29 proteins were expressed and purified and tested with 4 mixtures containing in total 55 metabolites. The protein-metabolite interactions were analyzed using NMR based methods. The authors report that they found 98 interactions with 76 being novel. However, only 30% of the known interactions could be verified and only 5 out of 8 novel interactions that the authors scored as allosteric were shown regulatory.

Furthermore the functionality of these interactions was found at extreme high concentrations of the interacting metabolites (e.g. 10mM).

Do the authors see the opportunity that such high concentrations can be achieved in vivo?

To clarify how *in vivo* concentrations relate to the concentrations used in our assay, we added an additional appendix figure (S10) highlighting the concentrations. Additionally, we inserted a statement in the third paragraph of the 'Discussion' section: "*In summary, of the five interactions showing significant effects on enzyme activity, two metabolites show the effect within their physiological concentration range (Zwf-GTP, FbaA-ATP), and three more show the effect at concentrations 1- to 7-fold higher than the anticipated physiological steady state concentration range of the metabolite (Zwf-ATP, FbaA-3PG, FbaA-PEP) (Park et al, 2016; Kochanowski et al, 2017) (Appendix Fig. S10).*"

We would respectfully want to point out though that physiological concentrations might be a somewhat misleading concept for at least three different reasons. Firstly, the reported values are only from a handful of conditions. Secondly, they are steady state concentrations when the cells are in homeostasis. Regulation, however, is expected to occur when cells are pushed out of homeostasis to help them find a new steady state, which is particularly true for the short-term metabolite-protein interactions tested here. It is well-known that dynamically changing concentrations may deviate far out of the steady state concentration ranges. Thirdly, the current estimates of cellular metabolite concentrations assume a uniform distribution of metabolites in cellular cytoplasm, which does not always hold true, as shown by some recent studies (Lohse 2017 Experimental and mathematical analysis of cAMP nanodomains). Since these consideration venture far beyond what we report, we prefer to not discuss them in the manuscript. Nevertheless, we fully agree that *in vivo* relevance is the now arising question from this work – actually also for the many already reported interactions – which goes far beyond *in vitro* enzyme assays. Our future work we will therefore follow-up in the suggested direction to assess *in vivo* functionality of some of the here detected interactions.

Why could the known interactions not be retrieved?

We would like to point out that by increasing the false-positive rate cutoff, retrieval of over 60% of known interactions could be achieved (see ROC curve in Appendix Figure S5). However, in order to minimize false-positive interactions, we opted for a lower false-positive, and consequently a lower true-positive rate. Additionally, we achieved a 2-fold higher true positive rate than a recent MS-based study (Piazza 2018, 16% recovery of known interactions). So it is not a question of being able to do it or not, but rather one of choice, where we went for smaller numbers at higher-confidence.

In this stage the study is very preliminary and does not lead to novel insights or regulatory principles, yet. The regulatory interactions seem at extreme concentrations, thus the relevance upon 'normal' metabolic regulation may be questioned.

If one would calculate the success of the method to measure the known interactions and the frequency that identified allosteric binders influence the activity of the target enzyme at extreme conditions the the publication is not mature at this stage.

Here, we respectfully disagree with the reviewer's opinion. Firstly, as described above, the discovered regulatory interactions do **not** occur at "extreme" concentrations but rather mostly within a range that could be expected to occur at least during dynamic adaptations, which is precisely the type of situation one would expect metabolite-protein regulation to be relevant. Secondly, we did not set out to identify novel regulatory principles, which in our view is in any case a tough call met only by very few papers. Our clearly defined goal was to probe the depth of our present knowledge of regulatory metabolite-protein interactions in the presently best-investigated biological subsystem. With this single piece of work, we essentially doubled the number of presently known interactions that resulted from literally hundreds to thousands of (wo)man years of hard work. While we are certainly aware of many of the limitations of our work – such as, for example, demonstrating the *in vivo* functionality for all these interactions – we strongly feel that we demonstrated the potential of the method and significantly contributed to completing our knowledge on the regulatory interaction topology in central metabolism. We hope that the reviewer can agree that although the manuscript does not answer all questions, it did achieve its stated goal and is a major step towards at least topological understanding.

Accepted

31st July 2019

Thank you again for sending us your revised manuscript. We are now satisfied with the modifications made and I am pleased to inform you that your paper has been accepted for publication.

Corresponding Author Name: Yaroslav Nikolaev, Uwe Sauer
 Manuscript Number: MSB-19-9008